# Beyond the Lab: What We Can Learn about Cancer from Wild and Domestic Animals

**DOI:** 10.3390/cancers14246177

**Published:** 2022-12-14

**Authors:** Hélène Schraverus, Yvan Larondelle, Melissa M. Page

**Affiliations:** Louvain Institute of Biomolecular Science and Technology (LIBST), UCLouvain, Croix du Sud 4-5/L7.07.03, B-1348 Louvain-la-Neuve, Belgium

**Keywords:** cancer research, alternative animal model, cancer resistance, cancer predisposition

## Abstract

**Simple Summary:**

In vivo cancer research primarily relies on rodent models, particularly transgenic mice. More recently, the doors have opened towards the use of unconventional species. Indeed, some of species presented in this review are characterized by a unique resistance to tumor development while others are prone to tumors that have also been detected in humans. We propose here a review of the mechanisms of resistance and tumor development present in these different species.

**Abstract:**

Cancer research has benefited immensely from the use of animal models. Several genetic tools accessible in rodent models have provided valuable insight into cellular and molecular mechanisms linked to cancer development or metastasis and various lines are available. However, at the same time, it is important to accompany these findings with those from alternative or non-model animals to offer new perspectives into the understanding of tumor development, prevention, and treatment. In this review, we first discuss animals characterized by little or no tumor development. Cancer incidence in small animals, such as the naked mole rat, blind mole rat and bats have been reported as almost negligible and tumor development may be inhibited by increased defense and repair mechanisms, altered cell cycle signaling and reduced rates of cell migration to avoid tumor microenvironments. On the other end of the size spectrum, large animals such as elephants and whales also appear to have low overall cancer rates, possibly due to gene replicates that are involved in apoptosis and therefore can inhibit uncontrolled cell cycle progression. While it is important to determine the mechanisms that lead to cancer protection in these animals, we can also take advantage of other animals that are highly susceptible to cancer, especially those which develop tumors similar to humans, such as carnivores or poultry. The use of such animals does not require the transplantation of malignant cancer cells or use of oncogenic substances as they spontaneously develop tumors of similar presentation and pathophysiology to those found in humans. For example, some tumor suppressor genes are highly conserved between humans and domestic species, and various tumors develop in similar ways or because of a common environment. These animals are therefore of great interest for broadening perspectives and techniques and for gathering information on the tumor mechanisms of certain types of cancer. Here we present a detailed review of alternative and/or non-model vertebrates, that can be used at different levels of cancer research to open new perspectives and fields of action.

## 1. Introduction

Cancer is responsible for one in six deaths worldwide and remains one of the leading causes of death in industrialized countries, second only to cardiovascular diseases [1]. Cancer development has been linked to several wide-ranging factors, including genetic pre-disposition or exposure to environmental carcinogens [2]. Cancerous cells can develop in almost every organ and have the ability to metastasize to other organs in the body, disrupting the functioning of biological systems through complex mechanisms that are still poorly understood. Many advancements have been achieved using in vitro and in vivo experimental approaches, the latter having heavily relied on the use of rodent models including genetically modified animals. These models have been valuable in our understanding of specific cancer-related issues. However, it is important to recognize the complementary wealth of knowledge that can arise by collaborating with researchers and/or veterinarians who work closely with domestic or wild animals, the latter being protected due to their habitation in zoos and/or animal sanctuaries. For example, by collecting data from a wide variety of domestic or wild animal species, it is observable that cancer incidence rate does not positively correlate with body size or lifespan [3]. Cancer risk is almost negligible in the African elephant or in the bowhead whale. Furthermore, humans have roughly one thousand-fold more cells than mice but are not a thousand times more likely to develop tumors. Preclinical and clinical observations demonstrate that approximately 50% of aged mice die of cancer compared to approximately 23% of aged humans [1,4,5,6]. Such differences in tumor occurrence between species are due to cancer development being cell specific as described by Peto’s paradox [3,7]. The purpose of this review is to synthesize information that has been collected from domestic and wild animals, including species that are either highly resistant or highly susceptible to cancer and in certain cases develop tumors that are similar to those in humans. This collection of data has the potential to aid in the development of precise and effective treatments and improve prognosis for cancer recovery in humans.

## 2. Limitations of Classical Experimental Models Used for Cancer Research

Numerous non-vertebrate organisms have been successfully and widely used as models to explore the different cellular and molecular mechanisms of cancer. The use of *Caenorhabditis elegans* and *Drosophila melanogaster*, for example, offers the advantage of well-established databases and molecular toolkits, in addition to a vast collection of well-defined mutant strain [8]. Rodents continue to represent the most common in vivo model and both mice and rats are routinely used to assess molecular mechanisms underlying cancer development and progression. The ease and ability to genetically modify mice, for example, allows us to address human-specific characteristics of cancer at the molecular and cellular levels as well as to perform surgical xenografts to explore questions linked to human-specific tumor development. Rodent models have successfully strengthened our understanding of cancer; however, their use is not without some limitations. In particular, transgenic mice which mimic conditions similar to human cancer, may also present with altered yet unrelated metabolic pathways that differ from the situation encountered in an animal that spontaneously develops cancer [9]. Along with environment and species physiological differences, the reliance solely on transgenic models may be reflected in the low success rates of clinical trials [10]. Therefore, it remains imperative that we incorporate results collected from other animals to provide us with novel insights into the development and progression of cancer. This can include animals that have already been introduced into laboratories as alternative animal models or domestic and wild animals in which tissues can be sampled under proper anesthetic procedures or at the post-mortem stage. Continuing to use model organisms while also collecting data from non-model organisms that are either cancer resistant or cancer prone, will help to identify novel molecular and cellular aspects underlying cancer development.

## 3. Potential Protective Mechanisms Responsible for Cancer-Resistant Vertebrates

Cancer resistance has been characterized in both small and large vertebrates. Specific pathways and mechanisms have been identified, which will be highlighted below. However, modifications to p53 signaling are often a common factor in cancer resistance. p53 is believed to act as a primordial tumor suppressor gene through its role as a transcription factor inducing the transcription of proteins involved in cell cycle regulation, apoptosis and DNA repair [5,11,12,13]. Although p53 is shared by all vertebrates, gene modifications that confer cancer resistance differ between species.

### 3.1. Naked Mole Rat

Living in a subterranean, eusocial setting, the average lifespan of naked mole rats (NMR; *Heterocephalus glaber*) often exceeds 20 years [14]. Several hundred autopsies have been performed on captive animals, and although environmental stressors differ between captive and wild settings, spontaneous neoplasms have not been identified. However, T cell lymphoma, testicular interstitial hyperplasia, adrenal hyperplasia and thymic atypical hyperplasia have been detected in a handful of individuals [15]. These findings suggest great genetic stability of their somatic cells that results in near-complete resistance against spontaneous tumor developmentt [16,17,18]. One potential explanation for the absence of tumors in NMR may be found in the differential expression of tumor suppressor genes, including Rb1 and p53 [19,20], which are involved in tumor prevention via their role in cell cycle regulation [5,13,21,22]. In experimental mice, if one of these tumor suppressors is inactivated during tumor development, uncontrolled cell proliferation continues, leading to tumor growth. In order to have tumor suppressor action in mice, both tumor suppressor proteins Rb1 and p53 as well as p19ARF must be inactivated. In NMR, a particular phenomenon occurs, in which a still undefined signal recognizes the inactivation of either p53 or Rb1 and the loss of one of these tumor suppressors causes apoptosis of the abnormally growing cells, leading to tumor growth arrest [19,20].

NMRs appear to have a large cache of species-specific cellular tools for defense against cancer development. For example, isolated NMR dermal fibroblasts undergo early contact inhibition (ECI), where cell growth arrest is triggered at a relatively low cell density compared to that of fibroblasts isolated from other mammals [19]. ECI appears to be regulated through the interaction between CD44 receptors and high-molecular mass hyaluronan (HMM-HA), which is a glycosaminoglycan produced in large quantities within NMR and blind mole rats (BMR; discussed below). This interaction activates the cyclin-dependent kinase inhibitor p16^INK4a^, or the NMR-specific pALT, which is a functional hybrid of p15^INK4b^ and p16^INK4a^ caused by alternative splicing [1]. Similar signaling has been characterized in vivo. Individually, p16^INK4a^ represses cell division by increasing the transition time from G1 to S phase, while p15^INK4b^ represses the overall progression of the cell cycle by inhibiting CDK4 and CDK6 activation. As a hybrid, pALT has a greater cell cycle inhibition capacity than p15^INK4b^ or p16^INK4a^ alone and ECI occurs even in cases where p16^INK4a^ is mutated or silenced [23].

Hyaluronan, isolated from a wide range of animals, possesses both pro- and anti-inflammatory and pro- and anti-proliferative properties. Cellular response appears to depend on the molecular weight and activation of such receptors as CD44, TLR2 and TLR4 [24]. NMRs possess a high molecular weight hyaluronan molecule (HMM-HA), which is secreted in higher amounts and appears to be more effective at preventing tumor progression compared to the hyaluronan secreted in mice and humans [19,24,25]. Tian et al. [26] demonstrated that HMM-HA inhibits tumor formation in mice xenografted with NMR fibroblasts, suggesting that HMM-HA has the potential to act as an effective tool in the control of cancer across species [27]. In contrast, evidence also indicates that hyaluronan levels are increased in tumor microenvironments contributing to cell proliferation and stem cell-like properties with enhanced immuno-suppressive properties that protect the tumor microenvironment [28]. Therefore, further research is required before HMM-HA can be used as a potential anti-cancer treatment in humans.

Other less defined mechanisms that may result in increased protection against tumor development include increased expression of the tumor suppressor Arf [20,29,30], which provides resistance against Ras oncogene activity [31]. Additionally, evidence suggests that differential microRNA (miR) expression can modulate tumor suppression via regulation of apoptosis, glycolysis, inflammation and mitochondrial metabolism [32]. Several lines of evidence also suggest that the activity of antioxidant enzymes, heat shock proteins and DNA repair enzymes are more active in NMRs compared to usual rodent models [33,34,35]. High retrotransposon activity has been reported in several human cancers [36]. Low levels and less efficient retrotranspose activity of the long interspersed nuclear element (LINE1) retrotransposon have been detected in the NMR genome, likely acting as a contributor to the reduced cancer incidence [37]. Increased L1 retrotransposition has been detected in several human cancers, yet has also been linked to promote cell senescence and has been shown to serve as a tumor suppressor at least within human myeloid leukemia [38]. NMR-LINE1 appears to induce cell senescence despite its low retrotranspose activity [38], suggesting other potential factors to determine whether retrotransposons function to induce or protect against cancer. Such protective and repair pathways have previously been demonstrated to combat against the development and/or progression of cancer but more work needs to be developed on these less defined mechanisms to provide more insight into enhanced cancer resistance in NMR [39,40,41].

### 3.2. Blind Mole Rat

Although more closely related to murids, blind mole rats (BMR; *Spalax spp*.) possess long lifespans for their body size, living upwards of 20 years [42]. Several cellular signaling mechanisms present in the BMR appear to be influenced by their subterranean living environment. They have developed a high tolerance to hypoxia and hypercapnia and are protected against hypoxia-induced apoptosis [1,43,44,45]. Point mutations within the p53 sequence cause amino acid substitutions that increase the transcription of genes related to cell cycle arrest while decreasing transcription of apoptotic genes [46,47]. Despite a reduction in expression of apoptotic genes, tumors have not been identified following autopsies of several hundred individuals, suggesting a resistance to tumor development [42]. BMRs are also less likely to develop cancer following treatment with the carcinogen 3-methylcolanthrene (3MCrA) compared to mice [48]. A comparison of the transcriptome profile between BMRs that developed tumors following treatment to those which did not, revealed differential expression of genes related to the extracellular matrix, cell cycle, and immune response. Whereas genes involved in DNA repair were differentially expressed in BMRs compared to mice [48]. BMRs possess anti-tumor mechanisms reported in NMRs that we have discussed above, such as over-secretion of HMM-HA, which here has been associated with an increased protection against ROS-induced damage [26,45]. In BMRs, HMM-HA, in addition to the presence of a unique dominant negative splicing variant of heparanase, appears to allow for improved structuring of the extracellular matrix and limits tumor growth and the development of metastases [49].

BMR fibroblasts do not display ECI as in NMR cells, however their cells do undergo concerted cell death (CCD) in response to hyperplasia, possibly due to increased secretion of interferon β (INFβ) at the site of hyperplasia [42,50]. This initiates apoptosis of premalignant hyperplastic cells via the p53 and Rb pathways and allows for early control of tumor development [42]. Activation of the INF pathway is typically induced by viral infection; however, several lines of evidence indicate that chromatin degradation and/or retrotransposon activity can also trigger this pathway [51,52]. Indeed, prior to the detection of CCD in BMR cells, the expression of several retrotransposon elements is elevated, supporting the role of retrotransposons in tumor suppression [50]. As alluded to above, the INF pathway may act as one potential factor involved in the role of retrotransposon activity in cancers. Evidence from several lung cancer cell lines reveal high levels of mutations within the INF pathway leading to the possible impairment of CCD [50]. BMRs may also be protected against tumor development due to the inhibition of the formation of the tumor microenvironment. This appears to be due to an upregulation of lactate clearance genes and a reduced migration of adipose tissue-derived stem cells (ADSC) [53,54]. Tumor microenvironments can be characterized by the presence of these mesenchymal cells, which support tumor growth and metastasis via various mechanisms including the promotion of angiogenesis. In BMRs, ADSCs appear to have a low capacity to migrate to the tumor microenvironment and therefore do not participate in its expansion or in tumor progression [54].

### 3.3. Bats

Accounting for nearly one-fifth of existent mammalian species, bats have been widely studied due to their ability for autonomous flight, their echolocation and their considerable longevity [55,56]. The oldest recorded individual was a Brandt’s bat (*Myotis Brandtii*) recaptured 41 years after first identification through banding [57]. The majority of bat research focuses on their role as a pathogen reservoir [58,59,60]. Several studies have also highlighted their low incidence of tumors, and are reportedly limited to leiomyosarcoma and sacromatid carcinoma [61,62,63], which may have arisen due to positive selective pressure on gene expression linked to mechanisms associated with flight [64,65,66]. For example, the expression of DNA repair and antioxidant enzymes is elevated in bats, to counteract the formation of ROS as a by-product of increased energy demands of flight. An upregulation of these enzymes may in parallel act as a mechanism that indirectly protects against tumorigenesis. Differences have been reported in DNA repair capacity via a positive selection for DNA repair enzymes, such as Ku80 that is involved in the repair of DNA double strand breaks [64]. Furthermore, insulin signaling is downregulated during periods of hibernation in bats, which has been linked to an overall reduction in cellular growth and proliferation in bats and other cancer resistant vertebrates [67,68,69,70,71].

Bats are viral reservoirs possessing unique viral tolerance, which may be partly due to expression of interferons (INF). For example, INFα3 induces the expression of several interferon-regulated genes (IRGs) that regulate DNA damage repair and/or inflammation, in the case of viral load [72]. Furthermore, approximately 27% of total IRGs identified in the black flying fox (*Pteropus alecto*) are unknown and have not been identified in other mammalian species. Pathway analysis indicates that most of the *P. alecto*-specific IRGs are enriched in cancer and organismal injury/abnormalities categories. These mechanisms, related to the potential viral load on these animals, may also indirectly protect against tumor development and progression [73]. Human viral proteins appear to upregulate the expression of ATP-binding cassette (ABC) transporters, although this has not been verified in bats, cells isolated from *P.alecto* contain significantly higher expression levels of the ABCB1 transporter compared to several human cell lines. This increased expression promotes protection against DNA damaging agents and may be one mechanism responsible for reduced cancer risk in bats [74].

Huang et al. [75] demonstrated that mouse-eared bats (*Myotis myotis*) display transcriptomic changes related to aging and tumor suppression mechanisms that vary from those observed in other mammals, including humans. For example, certain microRNAs have been identified as being differentially regulated in bats, as compared to model organisms, and are correlated to their increased longevity. These microRNAs include miR-101-3p, miR-16-5p, miR-143-3p and miR-155-5p that are upregulated and involved in tumor suppression by acting on mitotic cell cycle or DNA damage repair pathways. In contrast, selected microRNAs, such as miR-125-5p and miR-221-5pm, that promote tumor growth or whose gene targets are involved in regulating mitochondrial activity are downregulated [56,75].

Cancer resistance is not limited to only small mammals. In fact, tissue sampling from large and protected animals in zoos, wild-life reserves or in the wild, has allowed for the identification of other cellular and molecular pathways that appear to prevent cancer development in large land and aquatic mammals.

### 3.4. Elephant

Considering their mass, tumor incidence is lower than expected in elephants. As a result of retrotransposition, elephants possess retrogenes of p53 and LIF proteins that have been identified as potential key players in the protection against tumorigenesis in these large land mammals [76]. Fibroblasts isolated from African elephants (*Loxodonta Africana*) are resistant against genotoxic stress via amplification of the p53 locus, allowing the formation of 19 individual p53 retrogenes [5,13]. The mechanism of action of these retrogenes is not fully understood but it is possible that the proteins transcribed from the retrogenes bind to wildtype p53. This interaction therefore prevents wildtype p53 from binding with MDM2, a ligase which triggers p53 degradation by the proteasome. Conversely, the proteins of p53 retrogenes could bind directly to MDM2, inhibiting its activity [77]. In either case, the wildtype p53 protein would avoid degradation [13]. Modulation of p53 activity may sensitize cells to apoptosis induced by DNA damage [5]. Indeed, elephants appear to have developed an improved p53 response with greater apoptosis of fibroblasts than other species when exposed to carcinogens, such as doxorubicin or UV [13].

In addition to extra copies of p53, elephants also possess seven to eleven additional leukemia inhibitory factor (LIF) genes, whose protein product act as a pleiotropic cytokine by inhibiting differentiation on cell growth or by having a growth-promoting effect [78]. Although not all additional copies are expressed, due to the absence of regulatory elements, the LIF locus, in mammals, codes for three transcripts: LIF-D and LIF-M, which bind to the LIF receptor, and LIF-T, which is expressed in the cytoplasm and nucleus allowing caspase-dependent apoptosis [79]. In general, LIF acts as an extracellular cytokine binding to its receptor to activate phosphatidylinositol3kinase (PI3K), Janus kinase (JAK), signal transducer and activator of transcription 3 (stat3) and TGFβ signaling pathways, and prevent, as such, cell differentiation. In elephants, an additional copy transcript, LIF-6, is present and acts similarly to LIF-T but is localized in the mitochondria. Upon DNA damage, p53 upregulates LIF-6 transcription, which is targeted to the mitochondria to initiate cell apoptosis via Bax/Bak regulation [79]. Although much smaller in size, the North American beaver (*Castor canadensis*) appears to be another animal with several copies of genes that may be linked to tumor resistance, such as the tumor suppressor hydroprostaglandin dehydrogenase-15 (Hpgd) and aldehyde dehydrogenase 1 family member A1 (Aldh1a1), which encodes an enzyme responsible for lipid detoxification [80,81].

### 3.5. Whales

Whales are among the largest and longest-lived mammals on earth, with a record of 211 years for the bowhead whale *(Balaena mysticetus)* [82]. Similar to naked- and blind mole rats, whales are adapted to hypoxic conditions. In whales, this phenotype arises in part due to high levels of haemoglobin and myoglobin, therefore providing increased oxygen transport in red blood cells as well as increased oxygen cellular storage in the muscle tissue. The heart tissue of the whale is also likely protected against oxygen deprivation due to a sophisticated system that neutralizes the toxic build-up of nitrite [83,84].

Transcriptomic analysis of tissues sampled post mortem has revealed several unique amino acid substitutions and rapidly evolving genes in the bowhead whale compared to other species, including modifications to genes involved in insulin signaling, which have previously been reported to increase tumor resistance in bats and NMR [71,84,85]. Reduced signaling of the insulin/IGF1 pathway impairs regenerative capacity, which is a key component of tumor development [86]. In addition to the identification of a novel DNA repair enzyme transcript variant (Nei like DNA glycosylase 1; NEIL1) several genes that encode DNA repair and replication proteins contain amino acid substitutions. For example, amino acid substitution within excision repair cross complementation group 1 (ERCC1) and proliferating cell nuclear antigen (PCNA) correlate with higher expression levels [87]. Therefore, these phenomena may reduce DNA mutation rate and avoid tumor development [1,88].

A recent study published by Tejada-Martinez et al. [89] highlights the usefulness of cetaceans, in general, as an evolutionary model for studying resistance to tumor development. These authors demonstrated, by a comparative genomic approach, that cetaceans have at least seven positively selected tumor suppressor genes. Additionally, there is evidence for a gene gain and loss turnover more than twice as fast in cetaceans as in other mammals, as well as duplications of 71 genes. These duplicated genes are involved in DNA repair, glucose metabolism, and apoptosis, all of which provide significant protection against tumor development in these animals. This suggests not only an evolution towards animals with improved protection against tumors but also animals with efficient anti-aging processes.

The aquatic environment may play a role in the development of cancer. Spontaneous cancer rates are also relatively low in aquatic amphibians, possibly due to a specialized apoptotic pathway and immune system that must accommodate periods of metamorphosis and regeneration. In clawed- frogs (*Xenopus laevis*), the repair of damaged DNA occurs more slowly, which reinforces its error-free property. In addition, a direct-apoptosis process, independent of the cell cycle, allows the programming of cell death at any point in the cell cycle to avoid transformation into malignant cells [90].

### 3.6. Axolotls

Established laboratory lines of axolotls (*Ambyostoma mexicanum)* have provided insight into molecular and cellular mechanisms enabling tissue regeneration and lesion repair, two pathways linked to tumorigenesis [91,92]. As axolotls share many genes and signaling pathways with humans, they have the capacity to be a model for cancer therapy [93]. Axolotls have a particular life cycle as they are neotenic and can remain in the larval stage their entire life, while still reaching sexual maturity to successfully reproduce. This allows easy experimental manipulation of the individual larvae for research purposes [94]. Axolotls also have the ability for limb and organ regeneration. However, unlike in cancer, tissue growth occurs without fibrosis. Regeneration is initiated at the basal lamina of keratinocytes that migrate to cover the regeneration site. Within days, the new epithelium is innervated, and the apical epithelium cap is formed, where a suite of signaling molecules is produced to direct regrowth of both soft and hard tissues [95,96,97,98]. The regeneration process is almost error-free, partly due to the cell division signal, which cannot lead to cell proliferation with fibrosis. This complex process is associated with reprogramming genes of pluripotent adult cells and tumor suppressor genes, which are upregulated during tissue differentiation [99]. Interestingly, limb tissue extracts from axolotls induce cell cycle arrest and differentiation of human acute myeloid cells in culture, indicating anti-cancer potential of axolotl proteins [100]. Results suggest that if one of these processes is disrupted in the axolotl, uncontrolled cell proliferation and cancer incidence increases [94]. In addition to collecting data from axolotls, other amphibians and cold-blooded species in general can provide insight into cellular proliferation associated with cancer cells as well as the highly specialized microenvironment [94]. Above we described some unique tumor prevention mechanisms that have been characterized within alternative model organisms and summarized in Table 1.

Although there is currently little research on cancer development in reptiles, the limited data do suggest that they are not entirely protected against cancer, as some species of familiar lizard appear to develop a large amount of tumors, mostly on skin [101]. Therefore, data collected from these reptilian species can provide us with further insight into the development of cancer and possible targets for novel cancer therapy.

## 4. Non-Model and Alternative Model Vertebrate Species Susceptible to Cancer

Above, we focused on animal species that appear to have evolved molecular and cellular mechanisms that offer increased protection against cancer. However, there are animal species that readily develop cancer in ways similar to that observed within humans and can also be used to increase our understanding on the progression of this disease.

### 4.1. Carnivores

Tumors commonly identified in humans have also been detected in wild and domestic carnivores. For instance, mammary adenocarcinomas have been diagnosed in jaguars and wild dogs. Although not carnivores, similar tumors have been identified in Beluga whales [103]. This type of tumor is easily identified due to the visible tumor mass associated with the mammary tissue. In captive big cats, mammary tumors share similar presentation and clinical signs to those affecting humans. However, these tumors are often very aggressive with a poor prognosis [104]. Females of many big cat and domestic cat species are treated with progestogen contraceptives in attempts to promote conservation by allowing a return to reproduction when needed. However, such treatment has been linked to increased incidences of mammary tumors, especially within female jaguars, which can develop mammary tumors in a manner similar to humans carrying a BRCA1 genetic mutation that allows uncontrolled mammary cell proliferation [105].

Cancer is readily diagnosed in domestic and/or companion animals. Although this may be due to routine veterinary care, intensive breeding programs may also increase cancer predisposition in certain breeds of pets [106,107,108]. However, increased cancer rates may also occur due to the shared environment with humans, which may include unfavorable health conditions. A large portion of pets are overweight and have disrupted circadian rhythms, two factors that increase the risk of cancer in humans. In addition, there is evidence that secondhand cigarette smoke provokes similar symptoms within domesticated indoor pets, as in humans in terms of lung cancer [109].

Dogs can develop spontaneous tumors of similar clinical presentation and pathophysiology to humans and a number of spontaneously occurring cancer types, such as lymphoma, melanomas, osteosarcomas, bladder tumors, prostate, mammary tissue and lung cancers [110]. Similarly, chronic myeloid leukemia and bladder carcinomas are comparable in dogs and humans [111,112]. The degree of similarity shared between canine and human lymphomas has led to a better understanding of the disease in both species [113]. Oncogenes and tumor suppressor genes are highly conserved between dogs and humans, which at this level, suggests that data collected from dogs may offer us better insight into human cancer than those obtained from mice [114]. Domestic cats also offer many advantages in cancer research, and feline oral squamous cell carcinomas are clinically and molecularly similar to tumors of the head and neck in humans [109]. Mammary tumors, mostly aggressive and malignant in cats, are also commonly diagnosed in humans. Although the mechanisms of these forms of tumor are not fully understood, acute and/or chronic inflammation is a potential contributing factor [115,116,117]. Collaboration with veterinarians offers the possibility to evaluate cancer diagnosis, treatment and prevention methods in companion animals [110,118], which may provide further insight into the progression and treatment of human cancer. Similarly, ursids can develop a wide variety of cancers, ranging from mammary tumors in brown bears (*Ursus arctos*) to hepatic carcinomas in polar bears (*Ursus maritimus*). The panda and the sloths, which are mainly herbivorous, are not exempted [119,120].

### 4.2. Poultry

Several studies suggest that the epidemiological and molecular characteristics of spontaneous ovarian cancer in hens are similar to those observed in women. The prevalence of ovarian tumors in hens can exceed 35% depending on breed, age and egg production. Similarly, the formation of spontaneous tumors in 2-year-old laying hens suggests that this animal could be used to provide valuable insight into the development of human ovarian cancer. Certain histopathological subtypes and molecular characteristics have been identified in both hens and women. Genomic alterations involved in the development of ovarian cancer are similar between hens and women, for example, the mutation frequency of p53 and increased expression levels of CA-125 and E-cadherin have both been identified [121,122,123]. CA-125 and E-cadherin are currently used as biomarkers for detecting ovarian cancer in humans [124,125,126,127].

### 4.3. Killifish

Killifish have recently emerged as a novel aging model, due to their short lifespan that ranges between four to eight months in captivity [128,129,130,131]. In this species, embryos can enter diapause to avoid unfavorable environmental conditions. During diapause, embryonic metabolism is decreased to cease development and protect the embryo from desiccation as a normal environmental variation. As a result, embryonic development can vary from 17 days to three years. *Nothobranchius furzeri,* a strain with increasing ubiquity, has been reported to develop hepatic and renal carcinomas, as well as more rare neoplasms of the pancreas, gonads and swim bladder, with apparently higher rates in male individuals [128,132,133,134]. Although the occurrence of these carcinomas has recently been put into question [133,135], the use of killifish as an age-dependent model for tumorigenesis has the potential to greatly expand our knowledge into tissue-specific cancer development [136].

## 5. Discussion

The information presented above provides a picture, most likely partial, of little or unused resources in terms of animal models in cancer research. The information available and that still to come enable us to consider the concept of “one health” which integrates animal, human and ecosystem health to build an evolutionary map of global health. Indeed, some of the mechanisms presented above can be useful in terms of human health but also benefit veterinary medicine. For some of these features, a practical clinical application can be set up as, for example, in the case of LIF, a pleiotropic cytokine sensitizing tumor cells to immunotherapy. A wide variety of ligands target LIF receptors, opening the door to the use of small molecule therapeutics in both pets and humans [78]. In short, much remains to be done, but the use of multiple models, with characteristics directly found in humans or, on the contrary, totally unique, can only advance medicine, both human and veterinary. To this end, closer collaboration between universities, hospitals, veterinary clinics and zoos should be supported, allowing regular and efficient sampling, making the science malleable and accessible.

## 6. Conclusions

Much of our understanding of the development, progression and treatments of human cancers has been achieved by the use of animal models, such as mice that are often manipulated to mimic cancer events in humans. In this article, we provide examples of species which possess interesting aspects related to cancer development and/or progression, but which are often unused or underused in research. Many of the species presented display mechanisms related to longevity that can provide us with insights into how to reduce and/or prevent tumorigenesis. In contrast, several species are known to develop tumors similar to those present in humans. These systems are, therefore, of interest for a detailed understanding of the mechanisms of tumor development and progression, the implementation of precise diagnostic methods as well as various therapeutic trials. A more concerted and respectful use of these alternative species in cancer research, through an intensified collaboration between zoos, veterinary clinics and research groups would undoubtedly contribute to a better understanding of the mechanisms underlying cancer development and progression and provide new strategies for cancer prevention and treatment. We propose this range of species here to encourage collaboration between zoos, veterinary clinics, and research groups to collect data and samples in a complete, synchronized and respectful manner. We must keep an open mind to the richness that non-model organisms can offer us, not only in elucidating the mechanisms of cancer but potential cancer therapeutics. For example, a potential and novel breast cancer therapeutic may arise due to the anti-cancer properties identified within several strains of fungi collected from the fur of three-toed sloths (*Bradypus variegatus*) [137].

## Figures and Tables

**Table 1 cancers-14-06177-t001:** Species and their cellular mechanisms conferring protection against tumorigenesis.

Species	Cellular Mechanisms
Naked mole rats (captive populations)	Rapid apoptosis in case of p53, Rb1 or p19ARF loss [19]
	Early contact inhibition [19]
	High molecular mass hyaluronan [26]
	Differential microRNA expression [32]
	Tumor suppressor Arf increased expression [20]
	Elevated activity of antioxidant enzymes, heat shock proteins and DNA repair enzymes [33,34,35]
	Lower expression of genes involved in insulin/IGF1 signaling or GH signaling [71]
Blind mole rats (captive populations)	p53 mutation [47]
	High molecular mass hyaluronan [26,45]
	Heparanase [49]
	Concerted cell death [102]
	Adipose tissue stem cells low capacity of migration leading to a decrease of tumor microenvironment development [54]
Bats (wild and captive)	MicroRNA [56,75]
	DNA repair and antioxidant enzymes [65]
	Unique expression of INF [73]
	Downregulated insulin signaling [70]
Elephant (captive populations and frozen zoo *)	p53 retrogenes [5,13]
	3 LIF transcripts [79]
Whale (wild populations)	Downregulated insulin signaling [84]
	PCNA and ERCC1 increase [88]
	Duplication in 71 genes [89]
	Genetic turnover increase [89]
	At least 7 tumor suppressor genes [89]
Axolotl (captive populations)	Error free tissue regeneration and lesion repair [91,92]

* Conservation bank of various materials of animal origin such as DNA, oocytes, embryos or living tissues, stored at −196 °C.

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
