# Peer review of "Beyond the Lab: What We Can Learn about Cancer from Wild and Domestic Animals"

_cancers, 2022, doi:10.3390/cancers14246177_

Round 1
Reviewer 1 Report
In the submitted manuscript “Beyond the lab: What we can learn about cancer from wild and 2 domestic animals” the authors present an interesting view on the frequency of tumorigenesis in non-model animals, providing an evolutionary perspective on cancer in complex animals.
The traditional view of cancer has been on the dysregulated expression of one or more oncogenes, followed by the diversification of tumor cells during uncontrolled division. This altered expression can be due to the environmental factors or genetic predisposition of an individual. Despite the available knowledge and the different therapeutic approaches, a fundamental understanding of the origins of cancer is lacking. The submitted article addresses one such question by comparing cancer susceptibility in several wild and domesticated animals. The authors suggest that the extra copies or expression of genes regulating cell proliferation, DNA repair, or antioxidants have enabled the arrest of tumorigenesis in these animals. The presence of retrogenes (p53 in elephants) has also been proposed to scavenge MDM2 protein, thereby achieving stringent control over cell proliferation.
The article will help the readers understand the evolutionary strategies of different animals to evade cancer. Summarily, I recommend acceptance of the manuscript submitted to the Journal Cancers.
Author Response
We thank this reviewer for their interest in the review subject.
Reviewer 2 Report
The review entitled “Beyond the lab: What we can learn about cancer from wild and domestic animals” summarizes mechanisms of tumor resistance as well as those of tumor predisposition in different animal species. Due to the comprehensive nature of the topic, the authors provide selected examples for animal species with resistance to cancer and those with predisposition to certain neoplasms. The topic is of great value and interest for the scientific community involved in cancer research, prevention, and treatment.
The scientific value of the manuscript would be increased by the following additions:
- a brief discussion on how the presented information can be utilized in cancer research and its clinical application, i.e., cancer prevention, and treatment.
- illustrations summarizing the described data.
Comments and suggestions on the manuscript:
Keywords: please add “cancer predisposition”
- Line 40: please add “human beings”, so that the beginning of the sentence reads as follows: “
Cancer is responsible for one in six deaths in human beings worldwide….”
- Line 67: “Classical experimental models used for cancer research”, please replace “Limitations of classical experimental animal models used for cancer research”
3.1 Naked mole rat
- Lines 108-111. “In experimental mice, if one of these tumour suppressors is inactivated during tumour development, anarchic cell proliferation will continue, leading to tumour growth. In order to mount an anti-tumour response in mice, both these tumour suppressor proteins along with p19ARF must be inactivated. The anti-tumour response, if it occurs, is therefore delayed.
Could you please check the meaning of this paragraph. What is meant with “anti-tumour response”? How is the inactivation of tumor suppressor proteins performed and why does this suppress tumour growth? Should it not be expected that tumour growth will be enhanced in case tumour suppressor proteins are not functional?
- Lines 120-128: Please explain the connection between HMM-HA, CD44 and activation of the cyclin-dependent kinases p16INK4A and pALT. Please refer to the following reference: A. Seluanov, V. N. Gladyshev, J. Vijg, and V. Gorbunova, “Mechanisms of cancer resistance in long-lived mammals,” Nat. Rev. Cancer, vol. 18, no. 7, pp. 433–441, Jul. 2018, doi: 10.1038/s41568-018-0004-9.
- Lines 130-131: “Hyaluronan, isolated from a wide range of animals, is characterized as having anti-inflammatory, anti-metastasis, and anti-proliferative properties.” Please rephrase this sentence so convey the information, that 1) hyaluronan can have opposite functions, i.e. pro- or anti-inflammatory, pro- or anti-proliferative and that the type of functional activity is dependent on its molecular weight as well as its interaction with specific cellular receptors and 2) overexpression of hyaluronan can trigger tumorigenesis and in already existing tumors invasive behavior, proliferation of tumor cells as well as epithelial-mesenchymal transition.
- Lines 148-149: “In NMRs, differences have been reported in DNA repair capacity via a positive selection for DNA repair enzymes, such as Ku80 that is involved in the repair of DNA double strand breaks [33].” Could you please check if the cite reference is correct?
[33] G. Zhang et al., “Hypothalamic Programming of Systemic Aging Involving IKKβ/NF-κB and GnRH,” Nature, 481 vol. 497, no. 7448, pp. 211–216, May 2013, doi: 10.1038/nature12143.
- Lines 149-152: “More recently, results obtained from studies using NMRs suggested elevated levels of protein moonlighting, leading to increased activity of proteins involved in inflammation regulation [34].” Could you please briefly explain the mechanism of protein moonlighting that occurs in NMRs as well as the involved proteins and the correlation between regulation of inflammation and cancer development or progression.
3.2 Blind mole rat
- Please provide in the text the scientific name of the blind mole rat.
- Line 174: “Although BMR fibroblasts do not display ECI, they do undergo concerted cell death (CCD) in response to hyperplasia, possibly due to increased secretion of interferon β (INFβ) at the site of hyperplasia [45]. “ Could you please check, if this is the correct reference for the interferon beta secretion at the site of hyperplasia:
[45] Fang et al., “Adaptations to a Subterranean Environment and Longevity Revealed by the Analysis of Mole Rat Genomes,” Cell Rep., vol. 8, no. 5, pp. 1354–1364, Sep. 2014, doi: 10.1016/j.celrep.2014.07.030
3.3. Bats
- Lines 189-190: “However, several studies have highlighted their low incidence of tumours [51],[54].” Could you please check if this is the cited reference is correct. [54] G. Crameri et al., “Establishment, Immortalisation and Characterisation of Pteropid Bat Cell Lines,” PLOS ONE, 528 vol. 4, no. 12, p. e8266, Dec. 2009, doi: 10.1371/journal.pone.0008266.
- Lines 191-192: …..], which may have arisen due to positive selective pressure on gene expression linked to mechanisms associated with flight [33], [55], [56]. Could you please check if this is the correct citation at this location: [56] D. R. Valenzano et al., “The African Turquoise Killifish Genome Provides Insights into Evolution and Genetic Architecture of Lifespan,” Cell, vol. 163, no. 6, pp. 1539–1554, Dec. 2015, doi: 10.1016/j.cell.2015.11.008.
- Line 197: “other cancer resistance vertebrates”, please replace by “other cancer resistant vertebrates”.
- Lines 200-201: “One such INF” …Please provide the name of the interferon you are referring to.
- Line 202: “In addition, INF can induce”, Please provide the name of the type of interferon you are referring to.
- Lines 217-219: Please correct the following sentence: “In fact, issue sampling from large and protected animals in zoos, wild-life reserves or through careful sampling as with whales, has allowed for the..”
- Line 231: “the p53 retrogenes could bind directly to MDM2, inhibiting its activity” Do you mean “the proteins of p53 retrogenes” Please correct.
- Line 237: “LIF gene”, please provide first the non-abbreviated name and please write LIF in parenthesis, after having introduced this abbreviation, it can be used in the following text
- Lines 237-243: Please add to this paragraph the different partially opposite functions of LIF.
- Line 249: “Hpgd”, “Aldh1a1” Please provide for each protein first the non-abbreviated name and please write the abbreviations in parenthesis. After having introduced the abbreviation, it can be used in the following text
3.5 Whales
- Line 253: Please change “blink” to “blind”
- Lines 265 and 266: “ERCC1 (excision repair cross complementation group 1) and PCNA (proliferating cell nuclear antigen), please provide for each protein first the non-abbreviated name and please write the abbreviation in parenthesis. After having introduced the abbreviation, it can be used in the following text.
3.6. Axolotls
- Line 302: Could you please check if the following reference is correct [86] Y. Fradet, “Biomarkers in prostate cancer diagnosis and prognosis: Beyond prostate-specific antigen,” Curr. 602 Opin. Urol., vol. 19, no. 3, pp. 243–246, 2009, doi: 10.1097/MOU.0b013e32832a08b5.
4.1 Carnivores
- Lines 324: Please delete “Beluga whales” and reference 88. Please adjust the sentence and references according to the chapter “Carnivores”.
[88] D. Martineau et al., “Cancer in wildlife, a case study: beluga from the St. Lawrence estuary, Québec, Canada.,” 606 Environ. Health Perspect., vol. 110, no. 3, pp. 285–292, Mar. 2002.
- Lines 328-333. “Females of many big cat and domestic cat species are treated with progestogen contraceptives in attempts to promote conservation by allowing a return to reproduction when needed. However, such treatment has been linked to increased incidences of mammary tumours, especially within female jaguars, which can develop mammary tumours in a manner similar to humans carrying a BRCA1 genetic mutation that allows uncontrolled mammary cell proliferation [90], [91].”
Please replace reference [90] by a reference referring to the content of the sentences.
[90] S. W. Lee, C. L. Reimer, P. Oh, D. B. Campbell, and J. E. Schnitzer, “Tumour cell growth inhibition by caveolin re-expression in human breast cancer cells,” Oncogene, vol. 16, no. 11, Art. no. 11, Mar. 1998, doi: 612 10.1038/sj.onc.1201661
- Lines 339-341: “In addition, there is evidence second hand cigarette smoke provokes similar symptoms within domesticated indoor pets, as in humans in terms of lung cancer [95], [96].”
Please replace the cited references by those relating to the content of the sentence.
[95] L. D. MacDougall, “Mammary fibroadenomatous hyperplasia in a young cat attributed to treatment with 622 megestrol acetate,” Can. Vet. J., vol. 44, no. 3, pp. 227–229, Mar. 2003. 623
[96] A. P. Loretti, M. R. da Silva Ilha, J. Ordás, and J. M. de las Mulas, “Clinical, pathological and immunohistochem-624 ical study of feline mammary fibroepithelial hyperplasia following a single injection of depot medroxyprogesterone 625 acetate,” J. Feline Med. Surg., vol. 7, no. 1, pp. 43–52, Feb. 2005, doi: 10.1016/j.jfms.2004.05.002.
- Line 345: Please replace the wording “identical” by “comparable”
- Lines 348-350: “Oncogenes and tumour suppressor genes are highly conserved between dogs and humans, which at this level, suggests that data collected from dogs may offer us better in sight into human cancer than mice [101].” Please change the sentence as follows: “….than those obtained from mice”.
- Line 355: “potential candidate”, please replace by “potential contributing factor”
4.2. Poultry
Lines 364 and 369: please replace “female human” by “women”
- Lines 369-371: “The comparison of hen and human genomes has revealed notable similarities within the mutation frequency of p53 370 and in the increased expression levels of CA-125 and E-cadherin [109]–[111].”Please rewrite this sentence under consideration that it refers to the genomic alterations involved in the development of ovarian cancer and not to the entire genome.
Please add a brief discussion, that explains how the (described data obtained from “non-model” animal species) with cancer resistance or cancer predisposition can be utilized in cancer research and its clinical application, i.e. cancer prevention and treatment. Please also take the “one-health one-medicine” concept into consideration. It seems that the presented data could not only be beneficial for human medicine, but also for veterinary medicine.
Please consider the addition of illustrations to this article.
Citations in the text: Please correct the format of the references within the text according to the guidelines of the journal.
References: Please correct the format of the references in the reference list according to the guidelines of the journal. Please carefully check all references for completeness.
Table 1: “Species and their cellular mechanisms conferring protection against tumourigenesis”
- Naked mole rats “Lower expression of genes involved in insulin/IGF1 signalling or GH signalling [61]”
Could you please check if the cited reference is correct.
[61] S. Ma and V. N. Gladyshev, “Molecular signatures of longevity: Insights from cross-species comparative studies,” Semin. Cell Dev. Biol., vol. 70, pp. 190–203, Oct. 2017, doi: 10.1016/j.semcdb.2017.08.007.
- Whale:
„PCNA and ERCC1 increase [72]“
Could you please check if the cited reference is correct.
[72] B. A. Bluhm and R. Gradinger, “Regional Variability in Food Availability for Arctic Marine Mammals,” Ecol. 570 Appl., vol. 18, no. sp2, pp. S77–S96, 2008, doi: 10.1890/06-0562.1.
Author Response
Here are our answers to the comments. Thank you for your review.
Keywords: please add “cancer predisposition”
- Line 40: please add “human beings”, so that the beginning of the sentence reads as follows: “Cancer is responsible for one in six deaths in human beings worldwide….”
- Line 67: “Classical experimental models used for cancer research”, please replace “Limitations of classical experimental animal models used for cancer research”
- Line 197: “other cancer resistance vertebrates”, please replace by “other cancer resistant vertebrates”.
- Line 253: Please change “blink” to “blind”
- Line 345: Please replace the wording “identical” by “comparable”
- Lines 348-350: “Oncogenes and tumour suppressor genes are highly conserved between dogs and humans, which at this level, suggests that data collected from dogs may offer us better in sight into human cancer than mice [101].” Please change the sentence as follows: “….than those obtained from mice”.
- Line 355: “potential candidate”, please replace by “potential contributing factor”
- Lines 364 and 369: please replace “female human” by “women”
These have been corrected, thank you for drawing the mistakes to our attention.
Lines 108-111. “In experimental mice, if one of these tumour suppressors is inactivated during tumour development, anarchic cell proliferation will continue, leading to tumour growth. In order to mount an anti-tumour response in mice, both these tumour suppressor proteins along with p19ARF must be inactivated. The anti-tumour response, if it occurs, is therefore delayed. Could you please check the meaning of this paragraph. What is meant with “anti-tumour response”? How is the inactivation of tumor suppressor proteins performed and why does this suppress tumour growth? Should it not be expected that tumour growth will be enhanced in case tumour suppressor proteins are not functional?
Replaced by : In experimental mice, if one of these tumour suppressors is inactivated during tumour development, anarchic cell proliferation continues, leading to tumour growth. In order to have tumour suppressor action in mice, both tumour suppressor proteins Rb1 and p53 as well as p19ARF must be inactivated.
The paragraph have been rewritten to avoid misunderstanding. What we mean to say is that all the tumeur suppressors have to be activated to have and tumor suppression in mice
- Lines 120-128: Please explain the connection between HMM-HA, CD44 and activation of the cyclin-dependent kinases p16INK4A and pALT. Please refer to the following reference: A. Seluanov, V. N. Gladyshev, J. Vijg, and V. Gorbunova, “Mechanisms of cancer resistance in long-lived mammals,” Nat. Rev. Cancer, vol. 18, no. 7, pp. 433–441, Jul. 2018, doi: 10.1038/s41568-018-0004-9.
We have made the connection between HMM-HA, CD44 and the activation of the kinase inhibitors clearer, by rearranging the following sentences:
ECI appears to be regulated through the interaction between CD44 receptors and high-molecular mass hyaluronan (HMM-HA), which is a glycosaminoglycan produced in large quantities within NMR and blind mole rats (BMR; discussed below). This interaction activates the cyclin-dependent kinase inhibitor p16INK4a, or the NMR-specific pALT, which is a functional hybrid of p15INK4b and p16INK4a caused by alternative splicing (Add reference A. seluanov et al., Nat Rev Cancer 18:7, 2018).
- Lines 130-131: “Hyaluronan, isolated from a wide range of animals, is characterized as having anti-inflammatory, anti-metastasis, and anti-proliferative properties.” Please rephrase this sentence to convey the information, that 1) hyaluronan can have opposite functions, i.e. pro- or anti-inflammatory, pro- or anti-proliferative and that the type of functional activity is dependent on its molecular weight as well as its interaction with specific cellular receptors and 2) overexpression of hyaluronan can trigger tumorigenesis and in already existing tumors invasive behavior, proliferation of tumor cells as well as epithelial-mesenchymal transition.
Thank you for bringing the absence of this information to our attention. We have modified the text:
Hyaluronan, isolated from a wide range of animals, possesses both pro- and anti-inflammatory and pro- and anti-proliferative properties. Cellular response appears to depend on the molecular weight and activation of such receptors as CD44, TLR2 and TLR4 (Add A Tavianatou et al., Febs 2019).is characterised as having anti-inflammatory, anti-metastasis, and anti-proliferative properties. As such, hyaluronan production has been demonstrated to increase as a protective response to tumour development [23]. NMRs possess a very high molecular weight hyaluronan molecule, (vHMM-HA), which is secreted in higher amounts and appears to be more effective at preventing tumour progression compared to the hyaluronan secreted in mice and humans (Add A Tavianatou et al., Febs 2019) [18], [24]. Iian et al. [24] demonstrated that HMM-HA inhibits tumour formation in mice xenografted with NMR fibroblasts, suggesting that HMM-HA has the potential to act as an effective tool in the control of cancer across species [25]. In contrast, evidence also indicates that hyaluronan levels are increased in tumour microenvironments contributing to cell proliferation and stem cell-like properties with enhanced immuno-suppressive properties that protect the tumour microenvironment (add M Liu et al., Frontiers 2019). Therefore, further research is required before HMM-HA can be used as a potential anti-cancer treatment in humans.
- Lines 148-149: “In NMRs, differences have been reported in DNA repair capacity via a positive selection for DNA repair enzymes, such as Ku80 that is involved in the repair of DNA double strand breaks [33].” Could you please check if the cite reference is correct?
[33] G. Zhang et al., “Hypothalamic Programming of Systemic Aging Involving IKKβ/NF-κB and GnRH,” Nature, 481 vol. 497, no. 7448, pp. 211–216, May 2013, doi: 10.1038/nature12143.
the sentence and the reference have been moved to the paragraph about bats, where they were supposed to be from the beginning. Thank you for bringing this error to our attention.
- Lines 149-152: “More recently, results obtained from studies using NMRs suggested elevated levels of protein moonlighting, leading to increased activity of proteins involved in inflammation regulation [34].” Could you please briefly explain the mechanism of protein moonlighting that occurs in NMRs as well as the involved proteins and the correlation between regulation of inflammation and cancer development or progression.
Thank you for this recommendation, we have made the following changes to the text to provide more information on deamination and protein moonlighting. We also tried to link this section better with the previously mentioned sentences regarding a role for microRNAs.
MiR21 and miR155, with tumour suppressor and inflammatory targets are expressed in extracellular vesicles of NMRs, suggesting cell-to-cell transport and potential role in cellular communication (reference 34). In addition to miRs, several deiminated proteins were also identified within extracellular vesicles, including proteins involved in the HIF- (hypoxia inducible factor) signalling pathway and glycolysis as well as deiminated histones (reference 34). Protein deimination has been linked to structural and functional changes to proteins and is associated with potential moonlighting roles of proteins. Therefore, these targeted proteins and pathways may contribute to cancer resistance and longevity of the NMR.
- Please provide in the text the scientific name of the blind mole rat.
As there are several species of blind mole rat we have indicated that the genus is Spalax in the text.
- Line 174: “Although BMR fibroblasts do not display ECI, they do undergo concerted cell death (CCD) in response to hyperplasia, possibly due to increased secretion of interferon β (INFβ) at the site of hyperplasia [45]. “ Could you please check, if this is the correct reference for the interferon beta secretion at the site of hyperplasia:
[45] Fang et al., “Adaptations to a Subterranean Environment and Longevity Revealed by the Analysis of Mole Rat Genomes,” Cell Rep., vol. 8, no. 5, pp. 1354–1364, Sep. 2014, doi: 10.1016/j.celrep.2014.07.030
3.3. Bats
- Lines 189-190: “However, several studies have highlighted their low incidence of tumours [51],[54].” Could you please check if this is the cited reference is correct. [54] G. Crameri et al., “Establishment, Immortalisation and Characterisation of Pteropid Bat Cell Lines,” PLOS ONE, 528 vol. 4, no. 12, p. e8266, Dec. 2009, doi: 10.1371/journal.pone.0008266.
- Lines 191-192: …..], which may have arisen due to positive selective pressure on gene expression linked to mechanisms associated with flight [33], [55], [56]. Could you please check if this is the correct citation at this location: [56] D. R. Valenzano et al., “The African Turquoise Killifish Genome Provides Insights into Evolution and Genetic Architecture of Lifespan,” Cell, vol. 163, no. 6, pp. 1539–1554, Dec. 2015, doi: 10.1016/j.cell.2015.11.008.
These references correspond to the text.
- Lines 200-201: “One such INF” …Please provide the name of the interferon you are referring to.
- Line 202: “In addition, INF can induce”, Please provide the name of the type of interferon you are referring to.
We have clarified the two above requests and have re-structured the paragraph:
Bats are viral reservoirs possessing unique viral tolerance, which may be partly due to unique expression of interferons (INF). For example, INF-3 induces the expression of several interferon-regulated genes (IRGs) that regulate DNA damage repair and/or inflammation, in the case of viral load [62]. Furthermore, approximately 27% of total IRGs identified in the black flying fox (Pteropus alecto) are unknown and have not been identified in other mammalian species. Pathway analysis indicates that most of the P. alecto-specific IRGs are enriched in cancer and organismal injury/abnormalities categories. These mechanisms, related to the potential viral load on these animals, may also indirectly protect against tumour development and progression [63].
- Lines 217-219: Please correct the following sentence: “In fact, issue sampling from large and protected animals in zoos, wild-life reserves or through careful sampling as with whales, has allowed for the..”
We have replaced issue with tissue and also streamlined the sentence. It is now “In fact, tissue sampling from protected animals in zoos, wild-life reserves or in the wild, has allowed for the identification of other cellular and molecular pathways that appear to prevent cancer development in large land and aquatic mammals.
- Line 231: “the p53 retrogenes could bind directly to MDM2, inhibiting its activity” Do you mean “the proteins of p53 retrogenes” Please correct.
Yes, thank you. We have corrected this within the text.
- Line 237: “LIF gene”, please provide first the non-abbreviated name and please write LIF in parenthesis, after having introduced this abbreviation, it can be used in the following text
- Line 249: “Hpgd”, “Aldh1a1” Please provide for each protein first the non-abbreviated name and please write the abbreviations in parenthesis. After having introduced the abbreviation, it can be used in the following text
- Lines 265 and 266: “ERCC1 (excision repair cross complementation group 1) and PCNA (proliferating cell nuclear antigen), please provide for each protein first the non-abbreviated name and please write the abbreviation in parenthesis. After having introduced the abbreviation, it can be used in the following text.
These suggestions have all been addressed in the text.
- Lines 237-243: Please add to this paragraph the different partially opposite functions of LIF.
This have been adressed
4.1 Carnivores
- Lines 324: Please delete “Beluga whales” and reference 88. Please adjust the sentence and references according to the chapter “Carnivores”.
[88] D. Martineau et al., “Cancer in wildlife, a case study: beluga from the St. Lawrence estuary, Québec, Canada.,” 606 Environ. Health Perspect., vol. 110, no. 3, pp. 285–292, Mar. 2002.
Rather than delete this piece of information, we clearly state that these tumours have also been identified in beluga whales although they are not carnivores.
3.6. Axolotls
- Line 302: Could you please check if the following reference is correct [86] Y. Fradet, “Biomarkers in prostate cancer diagnosis and prognosis: Beyond prostate-specific antigen,” Curr. 602 Opin. Urol., vol. 19, no. 3, pp. 243–246, 2009, doi: 10.1097/MOU.0b013e32832a08b5.
- Lines 328-333. “Females of many big cat and domestic cat species are treated with progestogen contraceptives in attempts to promote conservation by allowing a return to reproduction when needed. However, such treatment has been linked to increased incidences of mammary tumours, especially within female jaguars, which can develop mammary tumours in a manner similar to humans carrying a BRCA1 genetic mutation that allows uncontrolled mammary cell proliferation [90], [91].”
Please replace reference [90] by a reference referring to the content of the sentences.
[90] S. W. Lee, C. L. Reimer, P. Oh, D. B. Campbell, and J. E. Schnitzer, “Tumour cell growth inhibition by caveolin re-expression in human breast cancer cells,” Oncogene, vol. 16, no. 11, Art. no. 11, Mar. 1998, doi: 612 10.1038/sj.onc.1201661
- Lines 339-341: “In addition, there is evidence second hand cigarette smoke provokes similar symptoms within domesticated indoor pets, as in humans in terms of lung cancer [95], [96].”
Please replace the cited references by those relating to the content of the sentence.
[95] L. D. MacDougall, “Mammary fibroadenomatous hyperplasia in a young cat attributed to treatment with 622 megestrol acetate,” Can. Vet. J., vol. 44, no. 3, pp. 227–229, Mar. 2003. 623
[96] A. P. Loretti, M. R. da Silva Ilha, J. Ordás, and J. M. de las Mulas, “Clinical, pathological and immunohistochem-624 ical study of feline mammary fibroepithelial hyperplasia following a single injection of depot medroxyprogesterone 625 acetate,” J. Feline Med. Surg., vol. 7, no. 1, pp. 43–52, Feb. 2005, doi: 10.1016/j.jfms.2004.05.002.
- Whale: „PCNA and ERCC1 increase [72]“
Could you please check if the cited reference is correct.
Table 1: “Species and their cellular mechanisms conferring protection against tumourigenesis”
- Naked mole rats “Lower expression of genes involved in insulin/IGF1 signalling or GH signalling [61]”
Could you please check if the cited reference is correct.
[61] S. Ma and V. N. Gladyshev, “Molecular signatures of longevity: Insights from cross-species comparative studies,” Semin. Cell Dev. Biol., vol. 70, pp. 190–203, Oct. 2017, doi: 10.1016/j.semcdb.2017.08.007.
We, somehow, faced a bibliography issue. This have been taken care of and all the references should be correct and complete.
- Lines 369-371: “The comparison of hen and human genomes has revealed notable similarities within the mutation frequency of p53 370 and in the increased expression levels of CA-125 and E-cadherin [109]–[111].”Please rewrite this sentence under consideration that it refers to the genomic alterations involved in the development of ovarian cancer and not to the entire genome.
We have clarified this point; this section is now as follows:
Genomic alterations involved in the development of ovarian cancer are similar between hens and women, for example, the mutation frequency of p53 and increased expression levels of CA-125 and E-cadherin have both been identified.
Thank you again for your careful reading of our work and your many comments.

Reviewer 3 Report
The review „Beyond the lab: What we can learn about cancer from wild and domestic animals” is focusing on a highly interesting topic, namely why certain species have a high/low risk to develop cancer.
After the introduction and a short paragraph on classical models for cancer, the third part of the manuscript expounds on species, which show low incidence of cancer and certain associated molecular characteristics of these species. This part seems written well and extensively, however, most references are older than three years. The last part of the manuscript is rather short and focusing on species susceptible to cancer. In my opinion, this is a very interesting topic worth to describe in more detail. Domestic animals are particularly interesting in this regard and might be an alternative or expansion to classical models in cancer research.
The main drawback of this manuscript is the reference list. Less than 20 percent of the references were published in the last five years. The references have to be updated.
Author Response
We thank this reviewer for their interest in the review subject. The references have been updated and some have been modified:
- Altwasser et al., 2019
- Bradford et al., 2010
- De Cecco et al., 2019
- Gu et al., 2021
- Halder et al., 2022
- Holm et al., 2022
- Koh et al., 2019
- Liu et al;? 2019
- McLelland et al., 2009
- Paradiya et al., 2022
- Siegal-Willott et al., 2007
- Suleiman et al., 2020
- Tavianatou et al., 2019
- Yamaguchi et al., 2021
- Zhang et al., 2020
- Zhao et al., 2021

Reviewer 4 Report
Dear Authors,
the work is interesting and well written, I have only a few suggestions to improve it.
- line 54 specify that they are unpublished data
-line 55 put the bibliography
-I really enjoyed Table 1 and would like one for non-model and alternative model vertebrates as well
- for non model and alternative model vertebrates I would like more insight into the genetic alterations in the various tumors. Example of dog and cat mammary tumors on molecular phenotypes, the dog is a good model for hormone-sensitive mammary carcinomas similar to women and the cat always in mammary tumors for non-hormone sensitive carcinomas similar to those of women. There are many things to tell for canine and feline oncology. the same investigations must be done for the other tumors of lina 344 and 345.
Author Response
We thank this reviewer for their interest in the review subject.
Dear Authors,
the work is interesting and well written, I have only a few suggestions to improve it.
- line 54 specify that they are unpublished data
-line 55 put the bibliography
Both lines are related to Tollis’ paper (2017). This have been added in the manuscript.
-I really enjoyed Table 1 and would like one for non-model and alternative model vertebrates as well
Non model and alternative models have been illustrated in table 1, such as whales and elephants.
- for non model and alternative model vertebrates I would like more insight into the genetic alterations in the various tumors. Example of dog and cat mammary tumors on molecular phenotypes, the dog is a good model for hormone-sensitive mammary carcinomas similar to women and the cat always in mammary tumors for non-hormone sensitive carcinomas similar to those of women. There are many things to tell for canine and feline oncology. the same investigations must be done for the other tumors of lina 344 and 345.
Thank you for your comment. Although we think that it would be interesting to list more tumours in domestic carnivores that are comparable to humans, this seems to us beyond the scope of this review, which is to provide the reader with a general overview.
Thank you again for your careful reading of our work.

Reviewer 5 Report
Schraverus and collegues describe in this review a series of spontaneous animal models among both domestic and wild animals. The article addresses tumor mechanisms in different species, offering a comprehensive analysis of cancer in animal models. I suggest some changes as follows:
Line 88: please define model versus non-model animals.
Line 103: please mention which types of tumors have been reported in naked mole rats
Line 109: uncontrolled is more appropriated than anarchic, please replace
Line 191: please mention which types of tumors have been reported in bats
Line 222: you have already mentioned that the incidence of tumors is not correlated with the size of the animal, therefore this sentence may be redundant.
Line 309: please discuss the histotypes; it adds interest to discuss tumor types.
Line 316: pets are spontaneous models for human neoplasms; spontaneous model would be a more precise definition than non-models or alternative models.
Line 342-358: considering the numerous studies of veterinary oncology and tumor ontogenesis, this part deserves to be better described, both regarding the genetic / molecular and protein alterations of specific tumors that are considered a spontaneous model for human tumors.
Line 353-355: it is very reductive to speak only of this mechanism considering that there are numerous studies on the molecular pathways of feline mammary carcinoma; please expand.
Line 404: I suggest reconsidering the definition of non-model by replacing it with spontaneous models.
Author Response
We thank this reviewer for their interest in the review subject.
Schraverus and collegues describe in this review a series of spontaneous animal models among both domestic and wild animals. The article addresses tumor mechanisms in different species, offering a comprehensive analysis of cancer in animal models. I suggest some changes as follows:
Line 88: please define model versus non-model animals.
The term non-model organism/animal is highly used in the literature and we do not believe it needs to be differentiated from model organism/animal; Nawy, 2011 (doi.org/10.1038/nmeth.1824); Ellegren, 2014 (doi.org/10.1016/j.tree.2013.09.008); Russel et al., 2017 (doi.org/10.1186/s12915-017-0391-5); Heck & Neely, 2021 (doi: 10.1021/acs.jproteome.0c00448).
Line 103: please mention which types of tumors have been reported in naked mole rats
We have corrected this to include that no spontaneous neoplasms have been detected but T cell lymphoma, testicular interstitial hyperplasia, adrenal hyperplasia and thymic atypical hyperplasia have been detected in a handful of individuals (Hadi et al., 2021).
Line 109: uncontrolled is more appropriated than anarchic, please replace
This has been corrected.
Line 191: please mention which types of tumors have been reported in bats.
The types of cancers identified and reported in the literature have been included in the text, leiomyosarcoma and sacromatid carcinoma.
Line 222: you have already mentioned that the incidence of tumors is not correlated with the size of the animal, therefore this sentence may be redundant.
Line 309: please discuss the histotypes; it adds interest to discuss tumor types.
Line 316: pets are spontaneous models for human neoplasms; spontaneous model would be a more precise definition than non-models or alternative models.
We have changed the heading to include spontaneous.
Line 342-358: considering the numerous studies of veterinary oncology and tumor ontogenesis, this part deserves to be better described, both regarding the genetic / molecular and protein alterations of specific tumors that are considered a spontaneous model for human tumors.
Line 353-355: it is very reductive to speak only of this mechanism considering that there are numerous studies on the molecular pathways of feline mammary carcinoma; please expand.
Line 404: I suggest reconsidering the definition of non-model by replacing it with spontaneous
We have kept non-model to describe the species that 1) may have enhanced protection against cancer or 2) present with spontaneous tumours. We understand that domestic animals may also routinely be used in research and therefore have renamed Section 4 to include spontaneous at there is more of a focus on domestic animals.
Thank you again for your careful reading of our work

Round 2
Reviewer 3 Report
Thanks for updating the references and including recent publications.